# The Use of Intravenous Fosfomycin in Clinical Practice: A 5-Year Retrospective Study in a Tertiary Hospital in Italy

**DOI:** 10.3390/antibiotics12060971

**Published:** 2023-05-27

**Authors:** Antonio Anastasia, Silvia Bonura, Raffaella Rubino, Giovanni Maurizio Giammanco, Irene Miccichè, Maria Rita Di Pace, Claudia Colomba, Antonio Cascio

**Affiliations:** 1Department of Health Promotion, Mother and Child Care, Internal Medicine and Medical Specialties “G D’Alessandro”, University of Palermo, 90127 Palermo, Italy; antonioanastasia90@gmail.com (A.A.); giovanni.giammanco@unipa.it (G.M.G.); mariarita.dipace@unipa.it (M.R.D.P.); claudia.colomba@unipa.it (C.C.); 2Infectious and Tropical Disease Unit and Sicilian Regional Reference Center for the Fight against AIDS, AOU Policlinico “P. Giaccone”, 90127 Palermo, Italy; silvia.bonura@policlinico.pa.it (S.B.); raffaella.rubino@policlinico.pa.it (R.R.); 3Antimicrobial Stewardship Team, AOU Policlinico “P. Giaccone”, 90127 Palermo, Italy; irene.micciche@policlinico.pa.it; 4Microbiology and Virology Unit, AOU Policlinico “P. Giaccone”, 90127 Palermo, Italy; 5UOC Farmacia, AOU Policlinico “P. Giaccone”, 90127 Palermo, Italy

**Keywords:** fosfomycin, antimicrobials, gram-negative, retrospective study

## Abstract

Fosfomycin in intravenous (IV) formulation has re-emerged as a valuable tool in the treatment of multi-drug resistant (MDR) and extensively drug-resistant (XDR) infections because of its broad spectrum of antibacterial action and pharmacokinetic characteristics. This retrospective study aimed to evaluate how fosfomycin was used in patients admitted to the Polyclinic of Palermo between January 2017 and July 2022. Clinical indications, therapeutic associations, clinical outcomes, and any side effects were analyzed. Intravenous fosfomycin was used in 343 patients, 63% male, with a mean age of 68 years (range 15–95). Urinary tract infections (UTIs) and hospital-acquired pneumonia (HAP) were the main indications for treatment (19% and 18% of the total cases, respectively), followed by skin and soft tissue infections and sepsis. IV fosfomycin was administered in combination with other antibacterial agents, the most common of which were ceftazidime/avibactam (35%), meropenem (17%), and colistin (14%). Nineteen patients received it as monotherapy for UTIs. About 66% had resolution of the infectious process with clinical remission (cure or discharge). Electrolyte disturbances occurred in 2.6% and gastrointestinal symptoms occurred in 2.9%. The data showed that IV fosfomycin is a safe and effective therapeutic option in the treatment of infections with multidrug-resistant microorganisms.

## 1. Introduction

The substantial decrease in the effective number of antibiotics and the slowdown in the development of new molecules have left few therapeutic tools for patients with multi-drug-resistant (MDR) or extensively drug-resistant (XDR) pathogen infections. From this point of view, the rediscovery of old antibiotics appeared to be one of the few viable therapeutic routes: fosfomycin proved useful due to its unique chemical structure and mechanism of action that does not exhibit cross-resistance with other antibiotics. Fosfomycin is a bactericidal antibiotic that interferes with cell wall synthesis, inhibiting UDP-N-acetylglucosamine enolpyruvyl transferase (MurA), an enzyme responsible for catalyzing the formation of N-acetylmuramic acid [1]. Gram-positive and Gram-negative bacteria require the formation of N-acetylmuramic acid for peptidoglycan synthesis, which means that fosfomycin’s spectrum of action is very broad [2,3]. Both time- and concentration-dependent activity have been suggested according to the bacteria evaluated, but due to its short half-life and rapid bactericidal action, a time-dependent approach is more often employed [4,5,6]. Specifically, it has been found to be highly active against *Staphylococcus aureus* and *Enterococcus* spp., and it exhibits considerable activity against Gram-negative bacteria such as *Escherichia coli* and *Klebsiella pneumoniae* including extended-spectrum beta-lactamases (ESBL) and carbapenemase producers [7,8,9]. Fosfomycin is generally used as part of a combination regimen and has been prescribed for various infections, such as complicated urinary tract infections (UTIs), osteomyelitis, skin and soft tissue infections, nosocomial lower respiratory tract infections, bacterial meningitis, and bacteremia/sepsis, in various countries [10,11]. This study aims to analyze the efficacy and safety of intravenous fosfomycin in the treatment of difficult-to-treat infections in a real-life setting.

## 2. Materials and Methods

This is a retrospective study conducted in the Hospital Paolo Giaccone in Palermo, Sicily, Italy. The records of all adult patients (> 18 years old) admitted to the infectious diseases ward or the different wards of the entire hospital, evaluated in consultancy, for whom fosfomycin in an intravenous formulation was administered for at least 24 h and each type of infection was analyzed. The period considered was between 1 January 2017 and 31 July 2022. Informed consent could not be obtained due to the study’s retrospective nature. 

A standardized Excel spreadsheet was used to collect and categorize information regarding the patients’ age, gender, clinical characteristics (comorbidities and type of infection), data related to fosfomycin use (empiric vs. target therapy and side effects), and outcomes. The dose of fosfomycin and the duration of treatment was decided by the specialist based on the type and severity of the infection and renal function, according to the manufacturer’s instructions. For patients with an estimated creatinine clearance (CrCl) of > 40 mL/min, a daily dose of 16–24 g/day divided into 3–4 doses was used. The daily dose was reduced by 20%, 40%, 60%, and 70% of the full dose in patients with CrCl of 31–40, 21–30, 11–20, and ≤ 10 mL/min, respectively. Fosfomycin susceptibility of the pathogens analyzed was interpreted using EUCAST cut-off points. The primary outcome was clinical response at the end of therapy. When the data were available, the patients were subdivided according to the resolution of the infectious process indicated as recovery if they achieved complete clinical success with resolution of signs/symptoms of infection, as relapse if after an initial improvement it was necessary to add an additional line of treatment, and as death if the subject died. As a secondary outcome, the occurrence of clinical- or laboratory-based adverse effects was assessed. Isolates and susceptibility testing were identified with the Vitek 2 system (bioMérieux, Lyon, France) or the broth microdilution method (BMD). The antibiotics used in the susceptibility testing for Gram-positive bacteria were glycopeptides, cephalosporins, penicillins, daptomycin, linezolid, aminoglycosides, fluoroquinolones, trimethoprim/sulfamethoxazole (TPM/SMX), clindamycin, and tetracyclines; the antibiotics used in the susceptibility testing for-Gram negative bacteria were penicillins, cephalosporins, carbapenems, aminoglycosides, fluoroquinolones, TPM/SMX, tetracyclines, and colistin. We obtained minimum inhibitory concentrations (MICs) for fosfomycin using the agar dilution method. The data were analyzed by descriptive statistics. Continuous variables were presented by the mean value and standard deviation (SD). Categorical variables were described by numbers and percentages.

## 3. Results

The population analyzed included 343 adult patients who received intravenous fosfomycin therapy for at least 24 h. The age of the patients ranged from 19 to 95 years, with a mean age of 68 years. Two-thirds were men (62.9%). As shown in Table 1, the background medical conditions included solid neoplasms (9.1%), cardiovascular disease (16.6%), pulmonary disease (16.6%), and diabetes mellitus (13.9%). Fifty-seven (16.6%) patients were admitted to a high-intensity care unit (ICU) at the time of starting IV fosfomycin therapy. In order of frequency, the main indications for fosfomycin administration were complicated and uncomplicated infections of the urinary tract (UTIs) (18.7%), lung infections (HAP) (18.4%), and skin and soft tissue infections (SSTI) including those involving surgical wound infections (15.5%) and bacteremia associated or not with signs and symptoms of sepsis (BSIs) (15.2%). IV fosfomycin has also been found to be indicated in the treatment of infectious diseases of the osteoarticular system and prosthesis, in the treatment of complicated endocarditis, infections of the central nervous system (CNS), and infections of various intrathoracic (mediastinitis, pleurisy, laryngeal fistulae, etc.) and intra-abdominal (IAIs) (cholecystitis, intra-abscesses, post-surgical intestinal infections, etc.) sites. All remaining infections that could not be categorized into the macro areas described above were designated as “other infections” (Figure 1). The indication for the use of fosfomycin was mostly based on the microbiological isolation of susceptible pathogens according to the EUCAST guidelines before the June 2022 amendment. Most of the infections detected involved only one pathogen (192) compared to polymicrobial infections (151).

In 57 patients (16.6%), no pathogen or other cause of infection was found, so empirical treatment was administered. Most of the bacteria isolated were Gram-negative (316 isolations), while Gram-positive bacteria were revealed in 60 cases. The most frequently encountered pathogens were in the order of frequency *Klebsiella pneumoniae* (193, 56.2%), *Pseudomonas aeruginosa* (42, 12.2%), *Acinetobacter baumannii* (36, 10.2%), *Staphylococcus aureus* (16, 4.6%), and *Enterococcus* spp. (28, 8.2%). The main microbiological isolates are shown in Figure 2 and completely described in Appendix A.

A sub-analysis of all *Klebsiella pneumoniae* isolates was also conducted aimed at analyzing the antibiotic resistance profile. Figure 3 shows the prevalence (%) of *K. pneumoniae* strains sensitive to the respective antibiotic classes over a total of 193 isolates. These data showed a high percentage of susceptibility of *K*.*pneumoniae* isolates to fosfomycin (92.2%) and colistin (75.6%). Figure 3, moreover, depicts the susceptibility of *K.pneumoniae* to ceftazidime/avibactam, ceftolozane/tazobactam, and meropenem/vaborbactam concerning the isolates assayed for the aforementioned antibiotics, which were subsequently introduced commercially. Figure 4 shows the prevalence (%) of *A. baumannii* strains sensitive to the respective classes of antibiotics in a total of 36 isolates, highlighting a total resistance to fluoroquinolones (levofloxacin and ciprofloxacin) and carbapenems and a high percentage of resistance to aminoglycosides (amikacin) and trimethoprim-sulfamethoxazole. The methods used for identification did not include the possibility of assessing the resistance profile of *A.baumannii* for fosfomycin and ceftazidime/avibactam, the latter often used because of the concomitant presence of infection with *K. pneumoniae* strains. Table 2 summarizes the number of recoveries, relapses, and deaths divided by the type of infections with their respective cases in ICUs and the main microbiological isolates.

In the majority (324, 94.4%), IV fosfomycin was used in combination with other antimicrobial agents. The main therapeutic partners were ceftazidime/avibactam (35,5%), meropenem (16.6%), colistin (14.1%), vancomycin (8.1%), and daptomycin (11.4%). 

The antibiotics most frequently used empirically in cases of infections without micro-biological findings were in the order of frequency meropenem (20 cases), daptomycin (14 cases), and vancomycin (8 cases). The use of monotherapy was chosen exclusively for the treatment of complicated and uncomplicated infections of the urinary system for a total of 19 patients. All antibiotic combinations used are described in Appendix A. The mean treatment duration was 12 ± 6.2 days, strongly depending on the indications and the main combination therapies. There was a resolution of the infectious process in 65.8% of cases. Patient death occurred in 26.2% of the total cases, while recurrence was observed in 4.9%. Comparisons of the various treatment indications and the corresponding cases and clinical resolution rates for each indication showed a success rate on several occasions of above 60 percent. Table 3 shows the number of recoveries, relapses, and deaths divided by the main combination therapies and specific isolates.

The drug safety analysis showed an absence of adverse events in 94.2%. The percentage of adverse events (5.8%) showed a prevalence of gastrointestinal symptoms (nausea and diarrhea) (2.9%), followed by isolated or multiple electrolyte disturbances (2.6%), and they occurred within the first week of treatment (Table 4). The need for discontinuation of treatment occurred only in four cases, due to uncorrectable hypokalemia. No serious adverse events were observed in the entire cohort analyzed. 

## 4. Discussion

The exponential increase in infections with MDR, XDR, or pandrug-resistant (PDR) microorganisms has rediscovered the therapeutic utility of old antibiotic molecules in monotherapy or in combination. Fosfomycin, among these, has emerged as being at the forefront of the treatment of such diseases. Our results on the use of intravenous fosfomycin in a population of 343 patients are in line with those of other studies in the literature. Fosfomycin has been classically used in UTIs but its use in other types of infections and against difficult-to-treat pathogens has gained more interest due to its unique mechanisms of action and favorable safety profile [10]. As already reported by Grabein et al. [10], the main indication for treatment in our case was a urinary tract infection (18.7%), including the use of monotherapy in that group of patients, with a rate of resolution greater than 70%.

The highest clinical success rates were observed in osteomyelitis (83.8%), SSTIs (67.3%), intra-abdominal infections (64.9%), and bacteremia/sepsis (61.5%), with success rates similar to those reported by other studies [12,13] but lower than those reported by Putensen et al. [14] (81%). On the other hand, resolution of the infectious process in the treatment of pneumonia and endocarditis appeared to be more limited in our cohort than in the literature studies [12,13], likely due to the severely compromised status of the patients and the concomitant presence of other diseases. This probably accounts for the high mortality rate shown in patients admitted to ICUs (85.9%) as shown in Table 2.

In the study of patients with documented infection, the main microbiological isolation was *K.pneumoniae* (56.2%). *K.pneumoniae* isolates were 82.3% sensitive to fosfomycin with MICs < 16 (76.6%) or equal to 32 (15.4%), and 82.4% were resistant to carbapenems. In all cases, IV fosfomycin was used in combination with other antibiotics with a resolution rate of 66.3%, which is in agreement with other studies in the literature [9,15,16,17]. These data, in addition to those we collected, seem to present IV fosfomycin as having valid therapeutic efficacy against *K. pneumoniae,* with various spectra of resistance proposing its use as a therapeutic alternative despite the absence of currently validated MIC breakpoints. Our use of the association of fosfomycin and ceftazidime/avibactam for *K.pneumoniae* and other Gram-negative bacteria obtained a clinical cure rate of 61.6%, similar to that described in other studies [18,19,20,21]. Worth reporting is the finding of *K.pneumoniae* isolates *bla*KPC resistant to ceftazidime/avibactam (10% of the isolates for which the antibiotic was assayed) and ceftolozane/tazobactam (92% of the isolates for which it was assayed), underlining the prevalence of pathogens resistive to new therapeutic molecules but sensitive to fosfomycin within our epidemiology. No resistance was observed to meropenem/vaborbactam. Regarding *S.aureus* and *E.coli* isolates, all those analyzed were sensitive to fosfomycin, confirming the finding already evaluated by Maraki et al. [22]. In our cohort, fosfomycin was used almost exclusively in combination with other antibiotics (94.4% of patients), supported by synergistic effects as recently recommended, especially if MDR Gram-negative pathogens are involved [13,23,24]. Remarkably, EUCAST stated that there are not enough data for providing any meaningful clinical breakpoint of fosfomycin against *P.aeruginosa* or *A.baumannii*, but our use in these types of infections was based on several clinical data on the efficacy of IV fosfomycin in combination with other antibiotics mainly derived from observational studies [25,26,27]. The most commonly used combination antibiotics were ceftazidime/avibactam, meropenem, and colistin with a resolution rate of 59.8% (73 of 122), 61.4% (35 of 57), and 51.1% (25 of 49), respectively. The main associations used are in line with the literature data based on both retrospective clinical trials and in vitro efficacy studies [3,10,15,26]. In agreement with Putensen et al. [14], also in our study, fosfomycin was not used as a therapeutic alternative in the perspective of a carbapenem-sparing regimen [28,29] but as a valid weapon of association with them especially in pneumonia (hospital-acquired and community-acquired pneumonia) and bacteremia/sepsis. Our use of the association of fosfomycin and colistin reported a clinical cure rate of 51.3%, similar to that described in other studies [26,30,31,32]. These studies and other studies in the literature analyzing combination therapy with ceftazidime/avibactam are described in Table 5. Although fosfomycin is not expected to be used in empirical treatment protocols, we used it in the treatment of particularly severe patients in relation to its synergistic potential and its broad spectrum of action, with a favorable outcome in 64.4% of cases. IV fosfomycin was found to be well tolerated and safe in almost all cases analyzed, with no occurrence of serious adverse effects. Only a small percentage of electrolyte disturbances (hypokalemia and/or hypernatremia) and gastrointestinal disturbances were found to be 2.6% and 2.9%, respectively, as already highlighted in other studies [10,24,33] and much less than that found in the cohort of Zirpe et al. [34]. However, this study does not remain free of limitations, ascertainable to its retrospective nature and the vast heterogeneity of the clinical setting of the patients for whom fosfomycin was used. Additionally noteworthy is the lack of a control group and the presence of co-administered antibiotics that act as confounding factors but are necessary for the treatment of infections with multidrug-resistant pathogens.

## 5. Conclusions

Our results suggest that fosfomycin is a safe and effective option for the treatment of difficult-to-treat infections due to Gram-negative and Gram-positive pathogens. It represents a valuable aid against extensively drug-resistant (XDR) and pan-drug-resistant (PDR) infections.

## Figures and Tables

**Figure 1 antibiotics-12-00971-f001:**
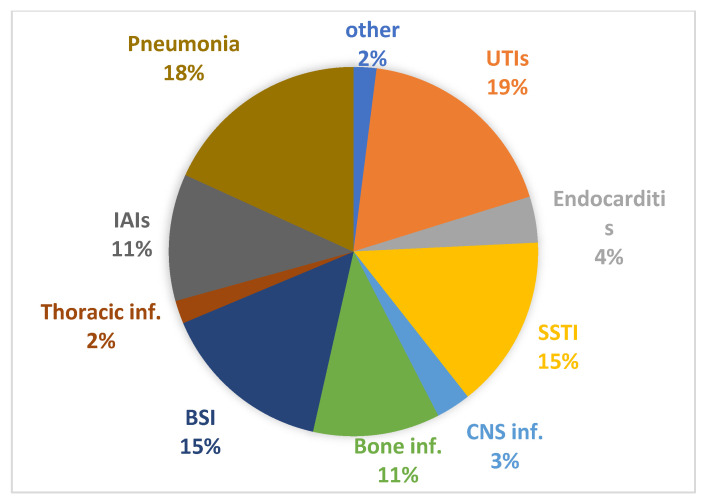
Indications for fosfomycin use. UTIs = urinary tract infections; IAIs = intra-abdominal infections; BSI = bloodstream infections; CNS inf. = central nervous system infections; SSTI = skin and soft tissue infections.

**Figure 2 antibiotics-12-00971-f002:**
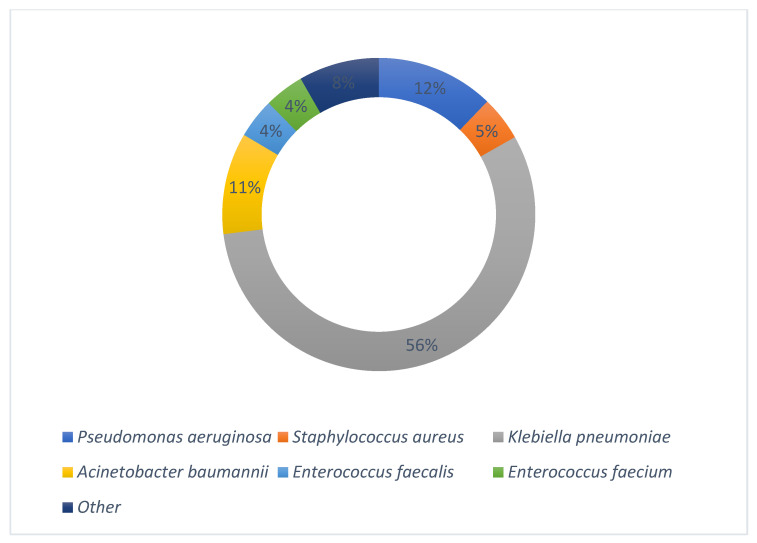
Main microbiological isolates.

**Figure 3 antibiotics-12-00971-f003:**
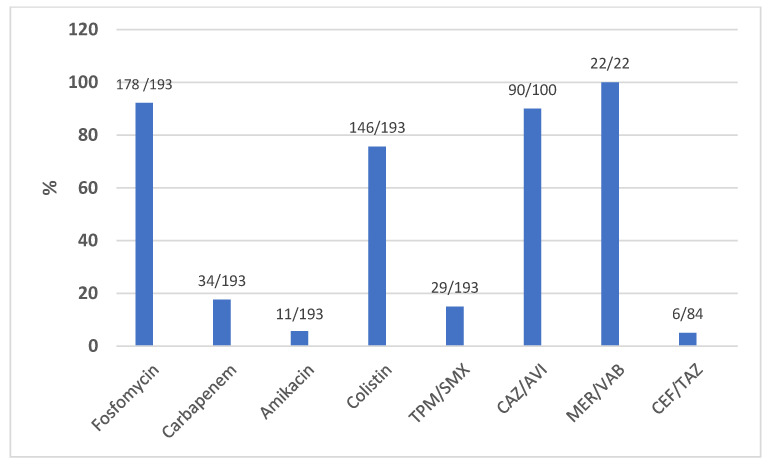
Prevalence of *K.pneumoniae* isolates sensitive to respective molecules in the extended antibiograms to the aforementioned antibiotics. Data shown as the percentages and number of susceptible strains/total isolates. TPM/SMX = trimethoprim–sulfamethoxazole CAZ/AVI = ceftazidime/avibactam MER/VAB = meropenem/vaborbactam CEF/TAZ = ceftolozane/tazobactam.

**Figure 4 antibiotics-12-00971-f004:**
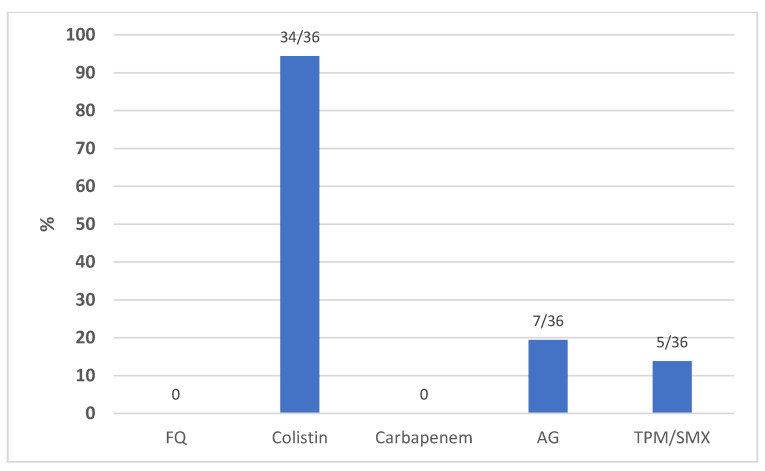
Prevalence of *A. baumannii* isolates sensitive to the respective molecules in the extended antibiograms to the aforementioned antibiotics. Data shown as the percentages and number of susceptible strains/total isolates. FQ = fluoroquinolones; AG = aminoglycoside; TPM/SMX = trimethoprim–sulfamethoxazole.

**Table 1 antibiotics-12-00971-t001:** The number of cases, recoveries, relapses, deaths, and missing data divided by the demographic characteristics and comorbidities.

	Cases *N*	Recovery *N* (%)	Relapse *N* (%)	Death *N* (%)	Missing Data
Total	343	226 (65.8)	17 (4.9)	90 (26.2)	10 (2.9)
Sex					
Male	216 (62.9)	148 (68.5)	9 (4.2)	54 (25)	5 (2.3)
Female	127 (37.1)	78 (61.4)	8 (6.2)	36 (28.3)	5 (3.9)
Age (mean ± DS) 68 ± 13.9 65.3 ± 13.9 64.1 ± 15 65.3 ± 14					
18–45 years	34	25 (73.5)	2 (5.9)	6 (17.6)	1
46–65 years	113	75 (66.4)	6 (5.3)	26 (23)	6
66–95 years	195	126 (64.6)	9 (4.6)	57 (29.2)	3
Comorbidity					
Solid neoplasm Hematological diseases Cardiovascular diseases Diabetes mellitus Kidney failure Lung diseases HIV/AIDS SARS-CoV-2 Other	31 (9.1) 29 (8.4) 57 (16.6) 48 (13.9) 29 (8.4) 57 (16.6) 11 (3.2) 18 (5.2) 59 (17.2)	21 (67.7) 18 (62.1) 35 (61.4) 37 (77.1) 20 (68.9) 29 (50.9) 9 (81.8) 9 (50) 38 (64.4)	3 (9.6) 1 (3.4) 3 (5.2) 1 (2.1) 2 (6.8) 4 (7) 0 2 (11.1) 3 (5.1)	4 (12.9) 9 (31) 16 (28.1) 9 (18.8) 7 (24.1) 23 (40.4) 2 (18.2) 7 (38.9) 17 (28.8)	3 1 3 1 - 1 - - 1

**Table 2 antibiotics-12-00971-t002:** Number of cases, recoveries, relapses, and deaths divided by the type of infection and the main microbiological isolates.

	Cases *N*	Recovery *N* (%)	Relapse *N* (%)	Death *N* (%)
Type of infection and subgroups in the ICU (intensive care unit)				
UTI/Pyelonephritis UTI/Pyelonephritis in the ICU Endocarditis Endocarditis in the ICU Skin and soft tissue infections (SSTI) SSTI in the ICU Central nervous system (CNS) infections CNS infections in the ICU Osteomyelitis Bacteremia/sepsis Bacteremia/sepsis in the ICU Intrathoracic infections Intrathoracic infections in the ICU Intra-abdominal infections Intra-abdominal infections in the ICU Pneumonia Pneumonia in the ICU Other infections	69 *6* 13 *4* 49 *4* 10 *4* 37 52 *12* 6 *2* 37 *9* 63 *16* 7	55 (79.7) *2 (40)* 6 (46.1) - 33 (67.3) - 6 (60) - 31 (83.8) 32 (61.5) *1 (8.3)* 3 (50) - 24 (64.9) - 31 (49.2) *3 (18.7)* 6 (85.7)	3 (4.3) - 2 (15.4) - 3 (6.1) - - - 2 (5.4) 1 (1.9) - 1 (10) - 1 (2.7) - 4 (6.3) - 0	11 (15.9) *4 (60)* 5 (38.5) *3 (75)* 10 (20.4) *3 (75)* 3 (30) *3 (75)* 2 (5.4) 18 (34.6) *11 (91.7)* 2 (40) *2 (100)* 12 (32.4) *9 (100)* 27 (42.9) *13 (81.3)* 1 (14.3)
**Main microbiological isolates**				
*Klebsiella pneumoniae*	193	129 (66.8)	5 (2.6)	57 (29.5)
*Pseudomonas aeruginosa*	42	28 (66.7)	1 (2.4)	7 (16.7)
*Acinetobacter baumannii*	36	17 (47.2)	2 (5.5)	16 (44.4)
*Staphylococcus aureus*	16	12 (75)	1 (6.3)	2 (12.5)
*Enterococcus spp.*	28	17 (60.7)	2 (7.1)	9 (32.1)

**Table 3 antibiotics-12-00971-t003:** The number of cases, recoveries, relapses, and death divided by the main combination therapies and specific isolates.

	Cases *N*	Recovery *N* (%)	Relapse *N* (%)	Death *N* (%)
**Main combination therapies and subgroups with specific isolates**				
MEROPENEM *Klebsiella pneumoniae * *Pseudomonas aeruginosa* *Acinetobacter baumannii* Empiric therapy	57 21 3 5 21	35 (61.4) 13 (61.9) 3 (100) 3 (60) 13 (61.9)	1 (1.8) - - - -	18 (31.5) 8 (38.1) - 2 (40) 8 (38.1)
CEFTAZIDIME–AVIBACTAM *Klebsiella pneumoniae* *Pseudomonas aeruginosa* *Acinetobacter baumannii* Empiric therapy	122 99 6 5 7	73 (59.8) 61 (61.6) - - 5 (71.4)	3 (2.5) 2 (2) - - -	42 (34.4) 35 (35.5) 6 (100) 5 (100) 2 (29.6)
COLISTIN *Klebsiella pneumoniae* *Pseudomonas aeruginosa* *Acinetobacter baumannii* Empiric therapy	49 23 6 17 3	25 (51.1) 13 (56.2) 4 (66.7) 8 (47.1) 1 (33.3)	2 (4.1) - - 2 (11.8) -	15 (30.6) 9 (39.1) - 7 (41.2) 2 (66.7)
DAPTOMYCIN *Staphylococcus aureus* *Enterococcus* spp Empiric therapy	39 4 3 21	30 (76.9) 4 (100) 2 (66.7) 14 (66.7)	3 (7.7) - - 3 (14.3)	6 (15.4) - 1 (33.1) 4 (19)
VANCOMYCIN *Staphylococcus aureus* *Enterococcus* spp Empiric therapy	28 2 3 9	20 (71.4) 1 (50) 1 (33.3) 7 (77.8)	- - - -	6 (21.4) - 1 (33.3) 2 (22.2)

**Table 4 antibiotics-12-00971-t004:** Safety of fosfomycin.

ADVERSE EVENT	*N* (%)
Absence of side effects Side effects ₋ nausea ₋ isolated hypernatremia ₋ isolated hypokalemia ₋ diarrhea ₋ rash ₋ hypernatremia and hypokalemia	323 (94.2) 20 (5.8) 7 (2) 4 (1.2) 4 (1.2) 3 (0.9) 1 (0.3) 1 (0.3)

**Table 5 antibiotics-12-00971-t005:** Comparison of the combination therapies and the clinical cure in our study and other studies.

**Combination Therapy with Intravenous Fosfomycin and Ceftazidime/Avibactam**				
**Author, year**	**Study**	**No. of cases**	**Pathogens**	**Clinical cure**
Our study	Retrospective	122	*K.pneumoniae (81.2 %) and other*	73/122 (59.8%)
Oliva et al., 2022 [18]	Retrospective, two-center	61	KPC-producing *K.pneumoniae*	45/61 (75.4%)
Zheng et al., 2021 [19]	Retrospective	6	Carbapenem-Resistant *K.pneumoniae*	4/6 (66.7%)
Burastero et al., 2022 [20]	Retrospective	9	*P.aeruginosa*	6/9 (66.7%) ¹
Zhanel et al., 2023 [21]	Retrospective	16	*K.pneumoniae*	11/16 (68.7%)
Gatti et al., 2022 [35]	Retrospective	2	*P.aeruginosa*	1/2 (50%) ¹
**Combination therapy with intravenous fosfomycin and colistin**				
**Author, year**	**Study**	**No. of cases**	**Pathogens**	**Clinical cure**
Our study	Retrospective	49	*K.pneumoniae, P.aeruginosa, A.baumannii*	25/49 (51.1%)
Apisarnthanarak et al., 2011 [30]	Retrospective	24	*P.aeruginosa*	14 (58%)
Sirijatuphat et al., 2014 [26]	Randomized controlled	47	*A.baumannii*	28 (59.6%)
Navarro et al., 2012 [31]	Prospective observational	4	*K.penumoniae*	3 (75%) ²
Assimakopoulos et al., 2023 [36]	Retrospective	7	*A.baumannii*	6/7 (85.7%) ¹
Russo et al., 2020 [27]	Prospective, observational	18	*A.baumannii*	16/18 (88.9%) ²
Thampithak et al., 2022 [32]	Retrospective	82	*A.baumannii, P.aeruginosa, Enterobacterales*	36/82 (43.9%) ²

^1^ The data in the study are described as “microbiological cure”; ^2^ the data in the study are indicated as “no death”.

## Data Availability

The data sets are available if required.

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
