# Peer review of "The Use of Intravenous Fosfomycin in Clinical Practice: A 5-Year Retrospective Study in a Tertiary Hospital in Italy"

_antibiotics, 2023, doi:10.3390/antibiotics12060971_

Round 1

Reviewer 1 Report

In the present manuscript, the authors analyzed the efficacy and safety of intravenous fosfomycin in the treatment of difficult-to-treat infections in a real-life setting. The article is interesting and well written. However, the authors must describe the procedures for the microbiologic isolation and antibiotic resistance profile in Material and methods.

Some minor comments:

Line 23:  “18% and 18% of total cases” change to “19% and 18% of total cases”.

Line 27: 67% or 66%?

Lines 28, 152, 208 and Table 4: please confirm the percentage of electrolyte disturbances and gastrointestinal symptoms.

Lines 35-36: define "MDR" and "XDR".

Lines 53-54: please rewrite the sentence and define “IV”.

Figure 1: please define the acronyms/abbreviations/initialism used.

Lines 113-114: “the latter being the most frequently encountered pathogen” This is redundant, please delete it.

Line 115: “Figure 3 analyzed…” Please change the verb. For example “showed”.

Figure 3: What aminoglycoside was used?

Line 141: 65.8% or 65.9% (see Table 1)?

Line 142: 4.9% or 4.6% (see Table 1)?

Line 144: Please change “almost always”. For example: “In several occasions”.

Table 2: add “ICU” in the second row of Type of infection.

Line 154: As a curiosity, what were the reasons for the suspension in these 4 cases?

Line 158: Define “PDR”.

Line 168: 67.9 or 67.3 (see Table 2)?

Line 182: Klebsiella spp. or K. pneumoniae?

Line 194: define “GN”.

Line 197: “51.1% (25 of 45)” or “25/49 (51.3%)” (see Table 5 and line 203)?

Line 202: define “CAP”.

Line 218: delete “infections”. 

References in the text: they must be formatted according to the journal guidelines.

Supplementary materials must be cited in the text.

An article that could be consulted and cited by the authors::

Zhanel G, Baxter M, Wong M, Mirzanejad Y, Lee A, Dhami R, Kosar J, Werry D, Irfan N, Tessier JF, Girourd G, Tascini C, von den Baumen TR, Walkty A, Karlowsky JA. Real-life experience with IV fosfomycin in Canada: Results from the Canadian LEadership on Antimicrobial Real-life usage (CLEAR) registry. J Glob Antimicrob Resist. 2023 Apr 6;33:171-176. doi: 10.1016/j.jgar.2023.03.010. Epub ahead of print. PMID: 37030573.

Minor editing of English language required.

Author Response

We are pleased to submit a revision of manuscript  ID: antibiotics-2372427 On behalf of all co-authors I would like to thank the reviewers for their thoughtful comments. We modified our manuscript based on their suggestions. Our point-by-point responses to their comments and corresponding modifications of our paper are the following ( probably some lines have changed due to the addition of tables and changes requested by reviewers)   In the present manuscript, the authors analyzed the efficacy and safety of intravenous fosfomycin in the treatment of difficult-to-treat infections in a real-life setting. The article is interesting and well written. However, the authors must describe the procedures for the microbiologic isolation and antibiotic resistance profile in Material and methods. (done in lines 86-88)   Some minor comments:   Line 23: “18% and 18% of total cases” change to “19% and 18% of total cases”.  (ok)   Line 27: 67% or 66%?  (66%)   Lines 28, 152, 208 and Table 4: please confirm the percentage of electrolyte disturbances and gastrointestinal symptoms.  (modified and concordant)   Lines 35-36: define "MDR" and "XDR".  (done)   Lines 53-54: please rewrite the sentence and define “IV”.   (done)   Figure 1: please define the acronyms/abbreviations/initialism used.  (done)   Lines 113-114: “the latter being the most frequently encountered pathogen” This is redundant, please delete it.  (deleted)   Line 115: “Figure 3 analyzed…” Please change the verb. For example “showed”.  (ok)   Figure 3: What aminoglycoside was used?  (We replaced the wording with amikacin as the only aminoglycoside used)   Line 141: 65.8% or 65.9% (see Table 1)?  (modified and concordant)   Line 142: 4.9% or 4.6% (see Table 1)?  (modified and concordant)   Line 144: Please change “almost always”. For example: “In several occasions”. (changed)   Table 2: add “ICU” in the second row of Type of infection. (Ok)   Line 154: As a curiosity, what were the reasons for the suspension in these 4 cases? (we specified the reason: uncorrectable hypokalaemia - line 193)   Line 158: Define “PDR”. (done - line 210)   Line 168: 67.9 or 67.3 (see Table 2)? (modified, now data concordant between table 2 and line 221)   Line 182: Klebsiella spp. or K. pneumoniae? (modified in K.pneumoniae)   Line 194: define “GN”. (rewritten as Gram negative line 248)   Line 197: “51.1% (25 of 45)” or “25/49 (51.3%)” (see Table 5 and line 203)? (Data are now modified and concordant)   Line 202: define “CAP”. (done)   Line 218: delete “infections”. (done)   References in the text: they must be formatted according to the journal guidelines. (Done according to the journal guidelines)   Supplementary materials must be cited in the text. (Done in lines 128 and 175)   An article that could be consulted and cited by the authors:: (DONE - the article is cited in table 5)   Zhanel G, Baxter M, Wong M, Mirzanejad Y, Lee A, Dhami R, Kosar J, Werry D, Irfan N, Tessier JF, Girourd G, Tascini C, von den Baumen TR, Walkty A, Karlowsky JA. Real-life experience with IV fosfomycin in Canada: Results from the Canadian LEadership on Antimicrobial Real-life usage (CLEAR) registry. J Glob Antimicrob Resist. 2023 Apr 6;33:171-176. doi: 10.1016/j.jgar.2023.03.010. Epub ahead of print. PMID: 37030573.      Comments on the Quality of English Language Minor editing of English language required. (done)   We would like to thank again the Editor and the reviewers for their instructive comments that substantially improved our paper. After revising the manuscript based on their suggestions, we hope that it is now acceptable for publication in Antibiotics. We would be glad to make additional modifications, if needed.    Sincerely, Antonio Cascio   Please see the attachment

Reviewer 2 Report

In a retrospective observationnal study tat included 343 patients, Antonio Anastasia et al. aimed to evaluate fosfomycin administration in clinical practice. Main indication were UTI and HAP and fosfomycin was mainly used in combination therapy with broad spectrum antimicrobial agent. Few indesirable effect were reported.

Overall, the study is of interest thanks to its correct sample size, and the quality of the data.

However, some issues needed to be discussed. I have 4 major comments and fews minors.

- First, Methods section need to be revise : definition of clinical response, which is the main endpoint, is not clear and have to be detail. Similarly, definition for relapse and main complicaiton have to be clarify.

-Second, because of the retrospective design, we may assume they were some missing data, but this is not reported in the tables. It is crucial to report number of missing data for each variables and how missingness was handled.

- Third, the authors mainly focused on Klebsiella pneumonia, which is the main micro-organims in their study. They report interesting data regarding antimicrobial suceptibility of KP but we miss similar data for other mirco-organisms. Especially, analysis of infections caused by acinetobacter baumanii may be interesting since mortality seems to be very high, especially in patients with co-amdinistration of ceftazidime-avibactam.

- I can not see any ethical consideration regarding this paper ?

Minor comments :

The introduction should be shortened, sentence regarding time depedent activity may be more appropriate in the discussion section. Conversely, data regarding frequently reported complications are awaited.

It is surprising that 57 patients whitout identified micro-organisms received Fosfomycin as empiric therapy, can you explain this unusual treatment in this population ?

Fig 3 : Could be expressed as % rather than raw numbers.

Finally, the low number of complicaitons is surprinsing, was there missing data regarding this outcome ? How were complications defined and screened ?

 Some minor corrections needed.

Author Response

We are pleased to submit a revision of manuscript  ID: antibiotics-2372427 On behalf of all co-authors I would like to thank the reviewers for their thoughtful comments. We modified our manuscript based on their suggestions. Our point-by-point responses to their comments and corresponding modifications of our paper are the following:   - First, Methods section need to be revise : definition of clinical response, which is the main endpoint, is not clear and have to be detail. Similarly, definition for relapse and main complicaiton have to be clarify. (done -  line 82-85)   -Second, because of the retrospective design, we may assume they were some missing data, but this is not reported in the tables. It is crucial to report number of missing data for each variables and how missingness was handled. (we have now reported the missing data in the modified Table 1. All the remaining data were collected using laboratory tests and medical records)   - Third, the authors mainly focused on Klebsiella pneumonia, which is the main micro-organims in their study. They report interesting data regarding antimicrobial suceptibility of KP but we miss similar data for other mirco-organisms. Especially, analysis of infections caused by acinetobacter baumanii may be interesting since mortality seems to be very high, especially in patients with co-amdinistration of ceftazidime-avibactam. (done - line 136-140; We added a new figure (Figure 4) with antimicrobial susceptibility of AB as requested)   - I can not see any ethical consideration regarding this paper ? (Ethical considerations and informed consent statement are described in lines 296 - 301)   Minor comments :   It is surprising that 57 patients whitout identified micro-organisms received Fosfomycin as empiric therapy, can you explain this unusual treatment in this population ? (as described in lines 259-262, we have used fosfomycin as empiric therapy in the treatment of particularly severe patients in relation to its synergistic potential and broad spectrum of action)   Fig 3 : Could be expressed as % rather than raw numbers. (We modified data, now shown as percentages on y-axis and number of susceptible strains/total isolates above the bars)   Finally, the low number of complicaitons is surprinsing, was there missing data regarding this outcome ? How were complications defined and screened ?  (there are no missing data about adverse effects/complications. We collected data using laboratory tests and medical records)   Please see the attachment

Round 2

Reviewer 1 Report

I thank the authors for their responses to my comments; however, there are still some minor comments that need to be addressed before accepting the manuscript:

Once the species are named, use the abbreviated genus in subsequent mentions.

Line 86-87: “Isolates were identified with the Vitek 2 system (bioMérieux) or the Broth microdilution method (BMD)”. BMD is used for antimicrobial susceptibility testing. Was the VITEK 2 system also used for susceptibility testing?

Antibiotics used in susceptibility testing should be mentioned in Material and methods.

Figure 4: please indicate the fluoroquinolones and aminoglycoside (amikacin) used. Discussion of this result is missing.

Minor editing of English language required

Author Response

Dear Editor, We are pleased to submit a revision of manuscript  ID: antibiotics-2372427 I would like to thank you for the thoughtful comments. We modified our manuscript based on  your suggestions. Our point-by-point responses to your comments (and corresponding modifications of our paper) are the following:  

Once the species are named, use the abbreviated genus in subsequent mentions. (done)

Line 86-87: “Isolates were identified with the Vitek 2 system (bioMérieux) or the Broth microdilution method (BMD)”. BMD is used for antimicrobial susceptibility testing. Was the VITEK 2 system also used for susceptibility testing? (yes, line 86)

Antibiotics used in susceptibility testing should be mentioned in Material and methods. (added in lines 87-92)

Figure 4: please indicate the fluoroquinolones and aminoglycoside (amikacin) used. Discussion of this result is missing. (antibiotics are now indicated in the figure, discussion was added in lines 143-145)

We would like to thank again the Editor and the reviewers for their instructive comments that substantially improved our paper. After revising the manuscript based on their suggestions, we hope that it is now acceptable for publication in Antibiotics. We would be glad to make additional modifications, if needed.    Sincerely, Antonio Cascio

Reviewer 2 Report

The authors performed the modifications suggested in the review.

No additional comments

Author Response

We would like to thank again the Editor and the reviewers for their instructive comments that substantially improved our paper. After revising the manuscript based on their suggestions, we hope that it is now acceptable for publication in Antibiotics. We would be glad to make additional modifications, if needed.    Sincerely, Antonio Cascio